# Automated Capability Evaluation of Foundation Models

## Abstract

Current evaluation frameworks for foundation models rely heavily on static, manually curated benchmarks, limiting their ability to capture the full breadth of model capabilities. This paper introduces Active learning for Capability Evaluation (ACE), a novel framework for scalable, automated, and fine-grained evaluation of foundation models. ACE leverages the knowledge embedded in powerful frontier models to decompose a domain into semantically meaningful capabilities and generates diverse evaluation tasks, significantly reducing human effort. In Mathematics, ACE generated 433 capabilities and 11,800 tasks, covering 94% of Wikipedia-defined skills in the domain while introducing novel, coherent ones. To maximize efficiency, ACE fits a *capability model* in latent semantic space, allowing reliable approximation of a subject model's performance by evaluating only a subset of capabilities via active learning. It reaches within 0.01 RMSE of exhaustive evaluation by evaluating less than half of capabilities. Compared to static datasets, ACE provides more balanced coverage and uncovers fine-grained differences that aggregate metrics fail to capture. Our results demonstrate that ACE provides a more complete and informative picture of model capabilities, which is essential for safe and well-informed deployment of foundation models.

## 1 Introduction

As foundation models grow in scale, generality, and influence across various domains, the challenge of understanding what they can and cannot do becomes increasingly urgent. Capability evaluations serve multiple purposes: they help practitioners select the right model for a given task, guide developers in improving model behavior, and most importantly ensure trustworthiness and safety, particularly in high-stakes domains such as cybersecurity, healthcare, and social engineering. Yet the current evaluation practices are dominated by static, human-curated benchmarks. While useful, these benchmarks quickly fall behind the pace of model development, missing fine-grained skills and introducing costly blind spots (Chen et al., 2021; Cobbe et al., 2021; Dua et al., 2019; Hendrycks et al., 2020; 2021b; Phan et al., 2025; Srivastava et al., 2022; Zellers et al., 2019; AI Security Institute, UK, 2024).

We argue that capability evaluation must itself become adaptive. Instead of freezing tasks in advance, one needs a process that can discover new capabilities as models evolve, generate meaningful and diverse tasks to probe them, and adaptively focus on the most informative regions of the capability space Zhang et al. (2024a); Prabhu et al. (2024). Recent advances in frontier and large language models (LLMs) make such adaptivity possible. LLMs can decompose a domain into semantically meaningful capabilities and generate diverse and contextually rich tasks for each capability. However, this power introduces a scalability problem: even a single domain may yield thousands of candidate capabilities, each requiring extensive task sets for reliable scoring. For commercial models (e.g., GPT-4, Claude, Gemini) with usage costs, exhaustive evaluation is prohibitively expensive.

We, therefore, propose to formulate capability evaluation as the problem of *approximating a latent capability function* where the goal is to estimate a model's competence across a large set of capabilities without exhaustively evaluating every one. The central research question becomes *how to approximate this function effectively when both the number of potential capabilities and the size of task sets required for reliable scoring are large.* Motivated by this question, we present **A**ctive Learning for **C**apability **E**valuation (ACE), a framework for automated, scalable, and fine-grained evaluation of foundation models. ACE operates in two stages: (1) it uses powerful frontier models to construct structured capability hierarchies and generate tasks with reference solutions; (2) it actively evaluates a subject model by learning its capability function in a latent space and selectively probing informative capabilities. The framework codebase is available at https://anonymous.4open.science/r/ace-7EAF for reproducibility and creating new evaluation benchmarks.

Our contributions are as follows:

- **Reframing capability evaluation.** We introduce ACE as the first framework that formulates evaluation as *approximating a latent capability function*, rather than exhaustively scoring on fixed benchmarks. We further show that the latent space, constructed via pretrained text encoders and dimensionality reduction, reliably preserves semantic relationships between capabilities, making principled generalization possible.
- **Adaptive coverage and efficiency.** By combining LLM-based capability decomposition with active learning in latent semantic space, ACE simultaneously expands coverage (capturing overlooked skills) and improves efficiency (minimizing evaluation cost). This resolves the long-standing trade-off between breadth and scalability in evaluation.
- **Large-scale empirical validation.** In Mathematics, spanning 433 capabilities and 11,800 tasks, ACE reveals capability- and area-level differences invisible to aggregate metrics. It recovers nearly the entire Wikipedia capability space, showing that automated benchmarks can surpass static, human-curated ones in coverage, granularity, and cost-effectiveness.

## 2 AUTOMATED CAPABILITY EVALUATION

### 2.1 PROBLEM STATEMENT

In our framework, we define a *capability* as an atomic skill or competence of the subject model (e.g., solving linear equations, factoring polynomials, or summarizing a passage). Capabilities are probed through *tasks*, each of which consists of a problem and a reference solution used for scoring. We formulate capability evaluation as the problem of approximating a latent *capability function* that reflects how well a model performs across a large space of candidate skills. Following Lu et al. (2025), the model under evaluation is referred to as the *subject model*. To construct the capability hierarchy and generate tasks autonomously, our framework relies on a set of frontier models with domain knowledge and reasoning ability. These models, collectively referred to as the *scientist models*, are responsible for proposing candidate capabilities, producing task instances, and supplying reference solutions for evaluation.

Let $\mathcal{C} = \{c_i\}_{i=1}^{N}$ denote the set of candidate capabilities produced by the scientist models. For evaluating a subject model, $\Omega$, each capability $c_i$ can be probed using an evaluation module $\texttt{Evaluate}(\cdot; \Omega)$ that generates multiple tasks, computes their outcomes, and returns an aggregated score $s_i \in \mathbb{R}_+$ for the subject model. Collectively, these scores define the latent capability function,

$$f_\Omega : \mathcal{C} \to \mathbb{R}_+, \quad \text{where,} \quad f_\Omega(c_i) = \mathbb{E}[\texttt{Evaluate}(c_i; \Omega)].$$

For simplicity, we omit $\Omega$ from the notation when it is clear from context. Obtaining a subject model's capability score via $\texttt{Evaluate}()$ is expensive as it requires designing, solving, and verifying tens of tasks.

Therefore, the objective is to approximate $f$ accurately while minimizing the number of calls to `Evaluate()`. This differs from static benchmarks, which predefine a fixed subset $\{c_i\}_{i=1}^m \subset \mathcal{C}$ and estimate $f$ by exhaustive evaluation on all tasks. Instead, we treat evaluation as an *active learning* problem in which we reliably estimate model performance across tasks while minimizing the number of capability evaluations by exploiting semantic relationships across $\mathcal{C}$.

Two challenges make this problem non-trivial:

- **Coverage**. The space of candidate capabilities is vast and open-ended; without principled generation, important skills may be missed.
- **Efficiency**. Even when capabilities are well-defined, an exhaustive evaluation could be expensive. A scalable framework must identify a small, informative subset that suffices to approximate the capability function reliably.

ACE addresses these challenges through two components: structured capability discovery, which organizes $\mathcal{C}$ into meaningful hierarchies, and latent modeling with active learning, which adaptively approximates $f$.

## 2.2 CAPABILITY HIERARCHY AND TASK DESIGN

**Capability Hierarchy.** Building on the definition of capabilities above, we next describe how they are organized and operationalized. At the top level, a domain, e.g., Mathematics, is divided into broad *areas* such as Algebra, Calculus, or Probability and Statistics; each area is then refined into specific *capabilities*, for example, Probability and Statistics is further broken down into capabilities such as Bayesian Inference, Markov Chain Probabilities, etc. This hierarchy is extensible and can support multiple levels of granularity depending on evaluation needs (Figure 1).

**Task Instantiation.** For each capability, the scientist models generate a set of *tasks*, each consisting of a problem and reference solution. Task formats are domain-dependent: in structured domains like Mathematics, problems usually admit unique solutions that can be deterministically verified; in open-ended domains such as summarization or scientific writing, multiple valid responses may exist, requiring more nuanced evaluation.

**Scoring.** The performance of a subject model on a task can be quantified either as a binary score, e.g., solved (1) vs. not solved (0), in domains with well-defined solutions, or as a continuous value in $[0, 1]$ to capture partial correctness or graded quality in open-ended domains. To obtain reliable capability-level estimates, the subject model is evaluated on a sufficiently large set of tasks through `Evaluate()`, which computes and aggregates task-level scores. By default, the mean is used; when tasks vary in difficulty or importance, weighted averages are applied, and in settings sensitive to outliers, the median is preferred.

**Verification.** Since the problem and reference solution for each task are generated automatically by the scientist models, we introduce a verification step. First, verification models review reference solutions for correctness. To further safeguard quality, we conduct targeted human inspection of outputs from both the task generation and verification stages (Appendix D). This ensures that the ground truth used for evaluation is reliable and reduces the risk of propagating errors during scoring. Second, subject model responses are evaluated against these references through `Evaluate()`. For structured tasks with deterministic answers (e.g., Mathematics), correctness is established through direct comparison. For close-ended solutions, exact match will be considered for evaluation. For open-ended tasks, we employ a judge model that scores responses against the task and reference solution on criteria such as accuracy, completeness, coherence, and relevance.

The judge model is provided with the task description, the reference solution, and the subject model's response. Judge prompts can be calibrated, and multiple judges can be ensembled to improve robustness. This layered approach ensures that both the ground-truth references and the subject model's outputs are evaluated rigorously and consistently. An abstract overview of our pipeline is provided in Figure 1 (Right).

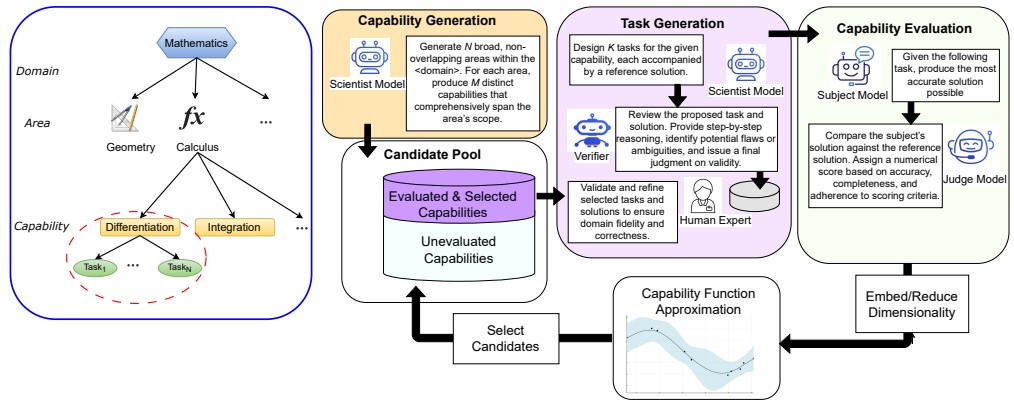

Figure 1: An overview of ACE. **Left:** Example capability hierarchy in Mathematics. **Right:** The ACE pipeline combining automated capability generation, task generation and verification, and active learning in latent space for efficient model evaluation.

## 2.3 LATENT MODELING OF CAPABILITIES

**Embedding.** We assume that capabilities in a domain are specified in a discrete space $\mathcal{T}$. For example, $\mathcal{T}$ could be the text space, where each capability is described by a short natural language statement. Direct function approximation in this space is challenging. We, therefore embed capabilities into a continuous latent space $\mathcal{Z} \subset \mathbb{R}^d$ using a pretrained encoder $E : \mathcal{T} \to \mathcal{Z}$. Each capability $t_i \in \mathcal{T}$ is mapped to $z_i = E(t_i)$, yielding dense semantic representations that support generalization across related capabilities. We assume the underlying capability function, $f$, is smooth in this space. This assumption is supported by empirical observations that LLMs exhibit correlated performance across related skills (Wang et al., 2024; Siska et al., 2024; Ilić & Gignac, 2024).

A key requirement of our approach is that semantically similar capabilities in $\mathcal{T}$ are mapped to nearby points in $\mathcal{Z}$. This property is essential for reliable generalization and uncertainty modeling. In Section 3.4, we empirically demonstrate that modern encoders satisfy this condition. Given an initial set of capability-score pairs $\{(t_i, s_i)\}_{i=1}^{N}$, with scores $s_i$ obtained from the subject model, the learning task reduces to approximating the capability function $f$ from the set of embedded pairs $\{(z_i, s_i)\}_{i=1}^{N}$.

**Function Approximation via Active Learning.** Exhaustive evaluation of all capabilities in $\mathcal{C}$ could be very expensive, as each call to Evaluate() involves generating, verifying, and scoring many tasks. To address this, we employ *active learning* to adaptively select informative capabilities. At each iteration, we compute the active learning *acquisition* scores[1] across unevaluated capabilities, select the optimal candidate, invoke Evaluate() on it, and add it to the training set for approximating the capability function. We then update the regression model. For regression, we adopt Gaussian Processes (GP), which provide both predictive means and uncertainty estimates, making it suitable for active learning (Malkomes, 2019; Gorissen et al., 2009; Fu, 2022; Riis et al., 2021; Chabanet et al., 2021). In our implementation we adopt the variance-reduction strategy of Cohn (1996), which selects the candidate capability that yields the largest expected reduction in posterior variance over the domain of $f$. This choice offers strong sample efficiency when evaluation budget is limited. Further details on GP regression and acquisition alternatives are provided in Appendix B.

---

[1]Acquisition function in active learning refers to the function used for selecting a candidate in each round.

**Dimensionality Reduction.** Capability embeddings are often high-dimensional, which, due to curse of dimensionality, can hinder GP regression. To address this, we apply dimensionality reduction $\varphi : \mathbb{R}^d \to \mathbb{R}^{d'}$ where $d' \ll d$, using methods such as Principal Component Analysis (PCA) or t-SNE (Van der Maaten & Hinton, 2008).

Bringing everything together, Algorithm 1 in Appendix A presents the full active learning procedure for capability function approximation.

## 3 EXPERIMENTS

### 3.1 SETUP

To assess the ACE framework, we focus on the domain of mathematics, which offers a hierarchical structure and well-defined problem formats, making it a natural testbed for capability-centric evaluation. In our experiments, we employ two scientist models, OpenAI `gpt-4o`[2] and Anthropic `Claude 3.7 Sonnet`[3].[4] The capability hierarchy is generated by the `gpt-4o` model, while tasks were generated by both models. To construct a diverse capability set, the scientist model is first prompted to propose broad and distinct areas within Mathematics. For each area, it is further prompted to produce specific capabilities in the modified `METR`[5] format following Lu et al. (2025). Each capability includes a `name`, `description`, and a corresponding `Python class`, which specifies exemplar tasks, task-solving instructions, and the scoring method. Full prompts for area- and capability-level generation are provided in Appendix F.1 and F.2.

These capabilities serve as input to the task generation pipeline. The pipeline begins by generating multiple diverse problems per capability. Each problem is then solved using the task-solving instructions specified in the capability's `Python class` to produce a solution. Together, the problem and solution form a complete task. We verify each task using a separate LLM call to confirm the correctness of the solution and to filter out incorrectly solved or unsolved tasks. Full task generation prompts are provided in Appendix F.3.

For evaluation, we use the `Inspect` framework (AI Security Institute, UK, 2024), which dynamically generates evaluation scripts based on the task-solving instructions and scoring method defined in each capability's `Python class`. For Mathematics, we adopt a binary scoring scheme: the subject model receives a score of 1 if its solution matches the reference solution, and 0 otherwise.

Following this procedure, we generate a benchmark of 433 distinct capabilities spanning 10 diverse mathematical areas. Although ACE does not require pre-generated task sets, for the purposes of experimentation and analysis we generated 11,800 tasks, with 27 tasks per capability on average. Experiments proceed in four stages: (i) coverage and task validity, (ii) capability benchmarking, (iii) validation of latent-space structure, and (iv) adaptive evaluation that approximates the capability function with active learning.

### 3.2 COVERAGE AND TASK VALIDITY

Our first question is *whether ACE-generated benchmarks provide comprehensive coverage and valid, discriminative tasks compared to established resources?* We compare three capability sets: (i) 287 ground-truth capabilities from Wikipedia (all sub-areas of mathematics from Wikipedia[6]), (ii) our ACE-generated synthetic benchmark, and (iii) Static human-curated math datasets, including MATH (Hendrycks et al., 2021b) and GSM8K (Cobbe et al., 2021).

---

[2]https://platform.openai.com/docs/models/gpt-4o
[3]https://www.anthropic.com/news/claude-3.7-sonnet
[4]At the time of this analysis these models were the frontier models of these companies.
[5]https://metr.org/
[6]https://en.wikipedia.org/wiki/Glossary_of_areas_of_mathematics

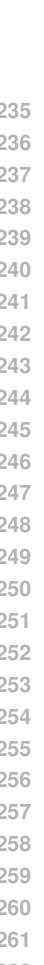

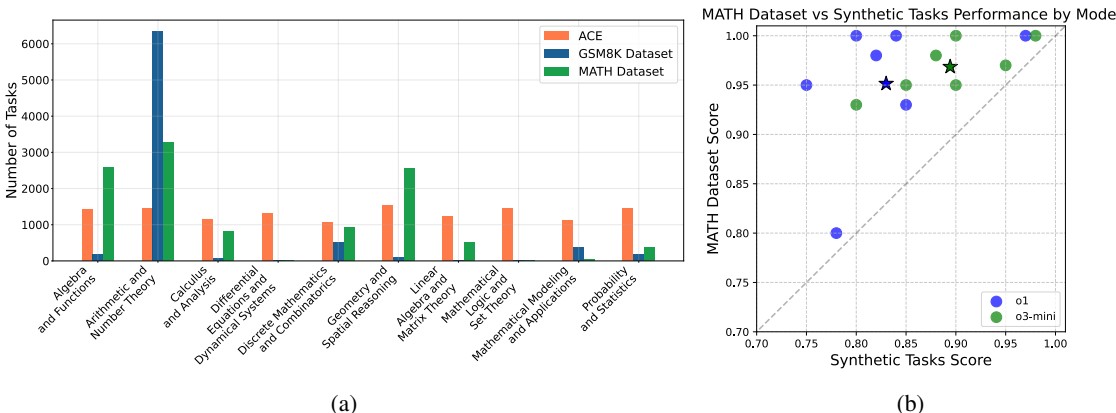

(a)                                                    (b)

Figure 2: Coverage and validity of ACE-generated benchmarks. (a) Task distributions across mathematical areas for ACE (orange), GSM8K dataset (blue) and MATH dataset (green). (b) Subject model performance on MATH vs. ACE (synthetic) tasks. Stars indicate average score across all capabilities.

**Wikipedia Coverage.** To quantify the overlap and differences between Wikipedia and ACE capability sets, we perform bidirectional matching analyses. Specifically, we use `Qwen2.5-32B-Instruct` with a classification prompt to map each capability from a source set to the most relevant capability in the target set. This constitutes a many-to-one matching problem: a given capability may map to a single best counterpart in the target set, but multiple source capabilities may map to the same target. Since many-to-one mappings are not symmetric, we conduct the analysis in both directions: Wikipedia → ACE and ACE → Wikipedia. Here, A → B indicates how many of capabilities in A are covered by B.

From the perspective of Wikipedia → ACE, 269 of 287 Wikipedia capabilities (94%) were matched to ACE capabilities, suggesting that ACE reliably captures nearly all widely recognized mathematical skills. From the perspective of ACE → Wikipedia, 405 of 433 ACE capabilities (93%) were covered by Wikipedia, while the remaining 28 appear to represent novel and potentially meaningful capabilities not explicitly covered in Wikipedia.

**Dataset Coverage.** To assess the coverage of static versus synthetic benchmarks, we categorize problems from MATH and GSM8K datasets into the high-level mathematical areas defined in ACE (see Section F.4 for details). The resulting distributions are shown in Figure 2(a). GSM8K exhibits a highly skewed distribution with a large fraction of tasks falling into the Arithmetic and Number Theory area, while other important areas (e.g., Differential Equations, Discrete Mathematics) are scarcely represented or entirely absent. The MATH dataset is less skewed than GSM8K, yet it lacks coverage in areas like Differential Equations. In contrast, ACE tasks are generated to achieve balanced coverage across all areas by design. This comparison highlights a key limitation of static benchmarks such as GSM8K and MATH: their task distributions often reflect dataset construction biases, leading to overrepresentation of certain skills and underrepresentation of others. Synthetic benchmarks like ACE mitigate this issue by enabling systematic and uniform coverage across the full capability space of the domain.

**Discriminative Power of Synthetic Benchmarks.** In this study we compare subject model scores on a synthetic benchmark and the MATH dataset. To construct the synthetic benchmark for each of the seven areas in MATH[7], we generate tasks using the scientist model. For each area, we then evaluate the performance of a subject model on synthetic tasks and the corresponding subset of MATH problems. We evaluate two subject

---

[7]Pre-algebra, Algebra, Number Theory, Counting and Probability, Geometry, Intermediate Algebra, and Pre-calculus

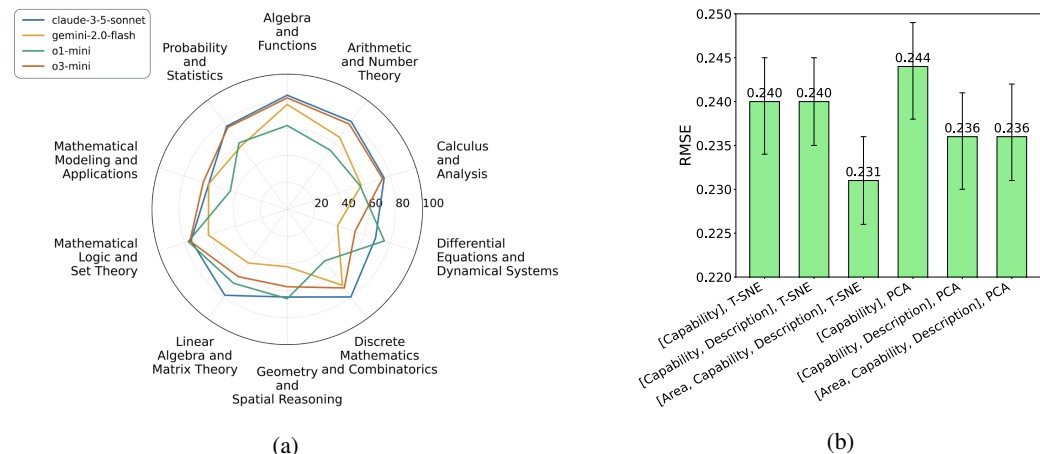

Figure 3: (a) Area-level benchmarking: subject model scores across mathematical areas. The reported score for each area is the average score of all capabilities within that area. (b) Semantic structure in latent space: Effect of input text and dimensionality reduction technique on capability function approximation.

models and show the results in Figure 2(b). For both subject models, comparing the distribution of scores across the two benchmarks reveals greater variation in scores on the synthetic benchmark. This indicates that tasks in the synthetic benchmark span a broader range of problem types and difficulties for each area. Consequently, this benchmark provides a more nuanced and discriminative assessment of model strengths and weaknesses.

### 3.3 FINE-GRAINED BENCHMARKING

Using ACE, we perform fine-grained evaluation of four subject models on all 433 capabilities of Mathematics. Area-level scores are computed by averaging capability-level scores. Figure 3(a) shows the results. Among these subject models Claude-3.5-Sonnet is the strongest and most consistent performer, maintaining high accuracy across nearly all areas. o3-mini follows closely. o1-mini performs well in Differential Equations and Dynamical Systems, but lags behind in several other areas. Finally, Gemini-2.0-Flash exhibits relatively low performance in areas such as Differential Equations or Calculus. These results illustrate the value of a structured fine-grained evaluation: even among generally strong models, there are differences in performance that may not be apparent in aggregate performance metrics.

### 3.4 SEMANTIC STRUCTURE IN LATENT SPACE

Reliable approximation of the capability function, $f$, depends on whether the latent space $\mathcal{Z}$ preserves semantic relationships between capabilities. In particular, capabilities within the same area should be embedded close to each other in $\mathcal{Z}$. Two components influence the structure of the latent space: the text encoder, which maps natural language descriptions of capabilities to high-dimensional embeddings, and the dimensionality reduction technique used to project these embeddings into a lower-dimensional space.

We first study the effect of the text encoder in isolation. We embed a subset of 20 capabilities sampled from 5 areas of Mathematics using the OpenAI text-embedding-3-small model[8](512-dimensional output).

---

[8]https://platform.openai.com/docs/guides/embeddings/

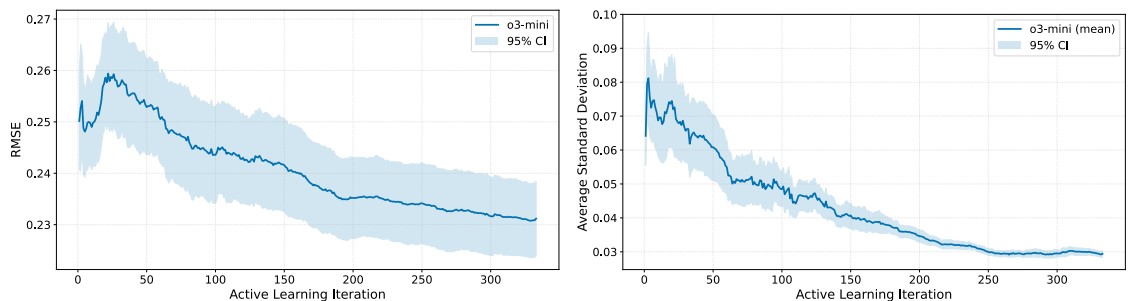

Figure 4: Performance of approximating the capability function. (Left) RMSE, (Right) Uncertainty (average standard deviation) over iterations of active learning. Shaded areas indicate 95% confidence intervals.

Each embedding is generated by concatenating the `area name`, `capability name`, and `capability description`. Pairwise cosine similarity analysis reveals clear intra-area clustering (Appendix C.2), indicating that embeddings capture meaningful semantic relations.

Next, we assess the combined effect of the text encoder and dimensionality reduction. We embed all 433 capabilities using the same encoder and project the resulting representations into a 2D latent space using t-SNE or PCA. Figure 5 in the Appendix shows the distribution of capabilities in the latent space. Both techniques preserve semantic relationships to varying degrees, but t-SNE produces more distinct clusters for capabilities within each area.

Finally, we assess how the encoder input choice and dimensionality reduction affect approximation of the capability function. Options for encoding a capability include (1) capability name, (2) capability name and description, and (3) capability name, description, and area name. Dimensionality reduction techniques consist of t-SNE and PCA. A Gaussian Process model is trained on 80% of the capability set, and Root Mean Square Error (RMSE) is reported on the test set. Figure 3(b) summarizes the results. We find that including richer input text (capability name, description, area name) and t-SNE yields the best performance.

### 3.5 ADAPTIVE EVALUATION FOR EFFICIENT APPROXIMATION

We conduct an ablation study of active learning in ACE to investigate the trade-off between efficiency and accuracy in function approximation. In practical model evaluation, the candidate pool for active learning would consist of all capabilities. For this experiment, however, we allocate 80% of the capabilities to the candidate pool and reserve the remaining 20% as a held-out test set for measuring generalization error. A GP model is initialized with two randomly selected capabilities from the training set and iteratively updated through active learning. At each iteration, the capability chosen is the one that yields the largest expected reduction in posterior variance over the domain of $f$ (Cohn, 1996) (see Appendix B). We use the scores of the `o3-mini` subject model for this study. Figure 4 presents RMSE (left), and predictive uncertainty (right) on the test set across active learning iterations. These results indicate that by evaluating the subject model on fewer than 50% of the capabilities (150 out of 346), the GP model achieves an RMSE within 0.01 of the target generalization error. Moreover, we observe a steady reduction in predictive uncertainty throughout the process.[9] These findings demonstrate that incorporating active learning in ACE provides effective generalization while substantially reducing evaluation cost.

---

[9]Additionally, Figure 7 in the Appendix shows area-level scores of the subject model when the capability function is fit on a fraction of capability scores. Note that these are reported scores only on the test set.

## 4 RELATED WORK

**Traditional Evaluation.** Early evaluation efforts relied on static, manually curated benchmarks such as MMLU (Hendrycks et al., 2021a), BIG-bench (Srivastava et al., 2022), and HELM (Liang et al., 2022), which aimed for broad coverage of general knowledge and reasoning. Other datasets target specific weaknesses, e.g., TruthfulQA (Lin et al., 2022) for factual reliability and ARC (Clark et al., 2018) for scientific reasoning. While influential, these benchmarks are inherently static, susceptible to contamination (Deng et al., 2024), and uneven across domains. Mathematics, for example, is relatively well served (e.g., GSM8K (Cobbe et al., 2021), MATH (Hendrycks et al., 2021b), but many applied and professional areas lack dedicated benchmarks. This motivates the need for adaptive frameworks that go beyond frozen datasets.

**Automated Evaluation.** Recent work leverages LLMs to generate, adapt, or score test cases, aiming to scale evaluation beyond fixed datasets. Model-assisted methods such as Dynabench (Kiela et al., 2021) and Adaptive Testing (Ribeiro & Lundberg, 2022) iteratively harden test sets. Structured approaches build task hierarchies (DARG (Zhang et al., 2024b), EvalTree (Zeng et al., 2025), TaskBench (Shen et al., 2024)), while autonomous systems such as AutoBencher (Li et al., 2025) and Automated Capability Discovery (Lu et al., 2025) aim for fully generative benchmarks. Other approaches optimize for particular objectives such as difficulty (Li et al., 2024), ethical reasoning (Jiang et al., 2025; Brown et al., 2025), or adversarial robustness (HarmBench (Mazeika et al., 2024)). These methods reveal important gaps overlooked by static resources but often remain constrained by predefined evaluation goals or reliance on existing benchmarks.

**Efficiency and Benchmark Optimization.** Benchmark generation is costly, and recent work explores active learning to target the most informative samples. Hassan et al. (Hassan et al., 2024) use clustering to expose rare, safety-critical cases, while Li et al. (Li et al., 2024) introduce RL-based subset selection for efficient evaluation. Despite these advances, many approaches remain tied to fixed datasets or optimize for narrow objectives, leaving open the broader challenge of discovering new capabilities and efficiently approximating performance across large and evolving skill spaces.

## 5 CONCLUSION

We introduced ACE, a framework for scalable, structured, and efficient evaluation of foundation models. ACE leverages frontier models to construct semantically meaningful capability hierarchies and associated evaluation tasks for a target domain. It further employs active learning in a latent semantic space to efficiently estimate a model's capability function and uncover strengths and weaknesses with minimal evaluation cost.

A limitation of the ACE framework is its reliance on frontier (scientist) models to generate, verify, and score tasks. While practical and scalable, this design raises valid questions about model hallucination, biases, and data contamination; however, employing several scientist models mitigate such risks to some extent. Designing a multi-agent framework where agents debate and critic each other's work could reduce some of these risks (Du et al., 2023; Liang et al., 2023). In addition, to estimate the true label (e.g., correctness of a generated solution), we can adopt statistical models such as the Dawid–Skene model (Dawid & Skene, 1979) or frameworks based on Item Response Theory (Baker, 2001), both of which are designed to aggregate noisy or uncertain judgments from multiple models. Adopting these techniques in the context of LLM-based evaluation is a promising direction for future work.

We believe ACE is a step toward scalable and adaptive evaluation of foundation models. As these models are increasingly deployed in high-stakes domains, the demand for fine-grained and cost-effective evaluation grows. By integrating frontier models with active learning, ACE lays the groundwork for rigorous and reliable evaluation.

## 6 ETHICS STATEMENT

ACE reduces human labor and improves scalability in capability evaluation, but it also relies on frontier "scientist models" to generate, verify, and score tasks. This design introduces risks of hallucination, bias in generated content, and potential data contamination from pretraining corpora. We mitigate these risks through multi-pass verification, targeted human inspection, and transparent reporting of limitations. Future extensions of ACE could incorporate multi-agent debate mechanisms or statistical aggregation methods (e.g., Dawid–Skene, Item Response Theory) to further improve robustness.

Our experiments are restricted to mathematics, where problems and solutions can be deterministically verified, minimizing risks of direct harm. While the framework could be extended to sensitive domains such as healthcare or law, such applications should proceed only with domain-specific oversight, ethical safeguards, and regulatory compliance (e.g., IRB approval, privacy protections).

By providing fine-grained, cost-efficient, and extensible evaluation, ACE can improve transparency and reliability in assessing foundation models, uncovering strengths and weaknesses that aggregate metrics overlook. However, uncritical adoption carries risks: benchmarks generated by ACE may inherit biases or errors from underlying models, and use in socially sensitive contexts without safeguards could exacerbate inequities. We emphasize that ACE should be applied responsibly, with human oversight and alignment to community standards on fairness, accountability, and transparency.

## 7 REPRODUCIBILITY STATEMENT

We have taken several steps to ensure the reproducibility of our work. The ACE framework source code, along with experimental configurations, training/test splits, acquisition functions, and analysis scripts, is open-sourced in our GitHub repository (https://anonymous.4open.science/r/ace-7EAF). A complete description of generated capabilities, tasks, evaluation procedures, and prompts is provided in the appendix and supplementary materials. In addition, we release JSON files containing all capabilities, areas, and scores produced by the scientist models, along with subject model scores and predictions from active learning. These resources make it possible for researchers to reproduce our benchmark construction end-to-end or to reuse any component of the framework independently.

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
