APPENDIX

# A    ACTIVE LEARNING FOR CAPABILITY FUNCTION APPROXIMATION

---

**Algorithm 1:** Active Learning for Capability Function Approximation

---
**Input:**
  Initial capability set $\mathcal{C} = \{c_i\}_{i=1}^N$ generated by the scientist models
  Pretrained encoder $E : \mathcal{C} \to \mathbb{R}^d$
  Dimensionality reduction method $\varphi$ (e.g., PCA, t-SNE)
  Evaluation module `Evaluate()` to score a capability
  Active learning acquisition function $\alpha(\cdot)$
  Target latent dimension $d' \ll d$

**Initialization:**
1. Encode all capabilities: $\mathbf{Z} = \{E(c_i)|c_i \in \mathcal{C}\}$
2. Reduce dimensionality: $\mathbf{Z}' = \varphi(\mathbf{Z}) \in \mathbb{R}^{N \times d'}$
3. Initialize training set $\mathcal{D}$ by randomly selecting a small number of capabilities (e.g., 2) from $\mathcal{C}$ and
 scoring them using `Evaluate()`

`// Active learning`
**while** *stopping conditions not met* **do**
  | 1. Fit GP model, $f$, on current $\mathcal{D}$ (non-parametric)
  | 2. Compute acquisition scores: $\forall \mathbf{z}_i' \in \mathbf{Z}' \setminus \mathcal{D}, \alpha_i \leftarrow \alpha(\mathbf{z}_i'; f)$
  | 3. Select the best candidate: $j \leftarrow \arg\max_i \alpha_i$
  | 4. Obtain capability score: $s_j \leftarrow \texttt{Evaluate}(c_j)$
  | 5. Update training set: $\mathcal{D} \leftarrow \mathcal{D} \cup \{(\mathbf{z}_j', s_j)\}$
**end**
**return** $\mathcal{D}$

---

# B    ACTIVE LEARNING WITH GAUSSIAN PROCESSES

A Gaussian process (GP) is a collection of random variables, any finite number of which have a joint Gaussian distribution (Rasmussen & Williams, 2006). It is fully specified by a mean function $m(\mathbf{x}) = \mathbb{E}[f(\mathbf{x})]$ and a covariance (kernel) function $k(\mathbf{x}, \mathbf{x}') = \mathbb{E}[(f(\mathbf{x}) - m(\mathbf{x}))(f(\mathbf{x}') - m(\mathbf{x}'))]$:

$$f(\mathbf{x}) \sim \mathcal{GP}(m(\mathbf{x}), k(\mathbf{x}, \mathbf{x}'))$$

Consider a regression task with training data $\mathcal{D} = \{(\mathbf{x}_i, y_i)\}_{i=1}^N$ where $y_i = f(\mathbf{x}_i) + \epsilon_i$ with $\epsilon_i \sim \mathcal{N}(0, \sigma_n^2)$. For a test input $\mathbf{x}_*$, the predictive distribution is Gaussian:

$$p(f_*|\mathbf{x}_*, \mathcal{D}) = \mathcal{N}(\mathbb{E}[f_*], \mathbb{V}[f_*]),$$

with predictive mean and variance:

$$\mathbb{E}[f_*] = {\mathbf{k}_*}^\top (\mathbf{K} + \sigma_n^2 \mathbf{I})^{-1} \mathbf{y} \tag{1}$$
$$\mathbb{V}[f_*] = k(\mathbf{x}_*, \mathbf{x}_*) - {\mathbf{k}_*}^\top (\mathbf{K} + \sigma_n^2 \mathbf{I})^{-1} \mathbf{k}_*, \tag{2}$$

in which $\mathbf{K}$ is the kernel matrix with $K_{ij} = k(\mathbf{x}_i, \mathbf{x}_j)$, $\mathbf{y} = \{y_1, \ldots, y_N\}$, and $\mathbf{k}_* = [k(\mathbf{x}_1, \mathbf{x}_*), ..., k(\mathbf{x}_N, \mathbf{x}_*)]^\top$.

The function-space view interprets the GP as defining a distribution over functions, where the kernel function encodes prior assumptions such as smoothness. A common choice is the squared exponential kernel:

$$k(\mathbf{x}, \mathbf{x}') = \sigma_f^2 \exp\left(-\frac{||\mathbf{x} - \mathbf{x}'||^2}{2l^2}\right).$$

GPs naturally lend themselves to active learning due to the availability of posterior mean and variance estimates. In particular two well-known approaches leverage GP posterior variance for active learning. MacKay (1992) aims at maximizing the expected information gain by selecting the data where the model has maximum variance. This is performed by selecting points that maximize the posterior variance:

$$\mathbf{x}^* = \arg\max_{\mathbf{x} \in \mathcal{U}} \mathbb{V}[f(\mathbf{x})], \tag{3}$$

where $\mathcal{U}$ is the pool of unlabeled candidates. This is equivalent to maximizing the reduction in entropy $H$ of the GP posterior:

$$\mathbf{x}^* = \arg\max_{\mathbf{x} \in \mathcal{U}} H[p(f|\mathcal{D})] - \mathbb{E}_{y|\mathbf{x}}[H[p(f|\mathcal{D} \cup (\mathbf{x}, y))]].$$

It is possible to perform optimization of Eq. 2 with respect to $\mathbf{x}^*$ using, e.g., gradient ascent (Seo et al., 2000).

The second method is motivated by minimizing the generalization error in terms of mean squared error (MSE). Using the bias-variance decomposition of MSE and making some assumptions with respect to the magnitude of bias, it can be shown that minimizing MSE can be approximated by choosing the candidate point that reduces the expected predictive variance over the entire input space (Cohn, 1996):

$$\mathbf{x}^* = \arg\min_{\mathbf{x} \in \mathcal{U}} \mathbb{E}_{y|\mathbf{x}}\left[\int \mathbb{V}[f(\mathbf{x}')|\mathcal{D} \cup (\mathbf{x}, y)]d\mathbf{x}'\right] \tag{4}$$

In practice the integration in Eq. 4 can be approximated by Monte Carlo or by calculating the variance over a holdout set.

For GPs, both approaches can be approximated efficiently as the posterior covariance matrix can be updated incrementally using rank-1 updates (Seeger, 2002). The active learning process iteratively fits the GP to current labeled data, $\mathcal{L}$, computes the acquisition score (Eq. 3 or 4) for all $\mathbf{x} \in \mathcal{U}$, selects $\mathbf{x}^*$ that maximizes the acquisition score, queries for $y^*$ at $\mathbf{x}^*$, and updates the labeled and candidate sets, $\mathcal{L} \leftarrow \mathcal{L} \cup \{(\mathbf{x}^*, y^*)\}$, $\mathcal{U} \leftarrow \mathcal{U} \setminus \{\mathbf{x}^*\}$.

## C    CAPABILITY DETAILS

In this section, we provide details on the generated capabilities, their embeddings used in our method, and LLM scores evaluated on each capability.

### C.1    CAPABILITY EMBEDDINGS

Figure 5 shows the distribution of capabilities in a latent 2D space for t-SNE (left) and PCA (right) dimensionality reduction techniques.

### C.2    CAPABILITY SIMILARITY HEATMAP

As expected, capability embeddings generated from capability names, areas, and descriptions carry the semantic similarity of the capabilities. Therefore, as the heatmap in Figure 6 shows, capability embeddings within the same area have higher cosine similarity compared to the capabilities in other areas.

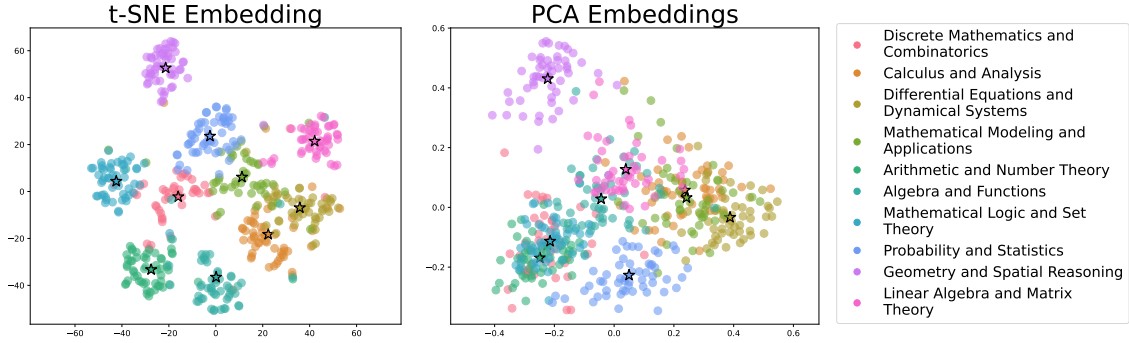

Figure 5: Two-dimensional representation of Mathematics capabilities using t-SNE (left) and PCA (right). Each point corresponds to a capability, and colors indicate high-level areas. Stars indicate the mean of capability representations for each area.

## D    MANUAL INSPECTION OF TASKS

To evaluate the quality of the task generation pipeline and the reliability of the automated verification step, we conducted a manual inspection of a subset of tasks. Specifically, we randomly selected 12 capabilities across three mathematical areas. For each capability, we sampled 15 tasks, resulting in a total of 180 problem–solution pairs. Each task's problem, solution, and verification model output were manually reviewed by solving the problem and comparing the correct solution to the automated verification outcome.

The results indicate a high degree of agreement between human and automated verification. Of the 180 tasks, we observed the following confusion matrix: True Positives = 158, False Negatives = 14, False Positives = 1, and True Negatives = 7. This corresponds to a precision of **99.4%**, recall of **91.9%**, and overall verification accuracy of **91.7%**. These results support the conclusion that the automated pipeline for task generation and verification is reliable for evaluating model capabilities at scale. Despite strong performance in task generation, our inspection surfaced a few recurring issues that are important to address in future iterations of the framework:

1. **Rounding Errors.** Infrequent but notable rounding inaccuracies occurred when intermediate numerical results were used in subsequent calculations. These rounding issues sometimes led to small deviations in final answers and highlight the need for improved numerical precision handling.

2. **Lack of Task Diversity.** Many tasks within a capability were structurally or conceptually similar. Increasing task diversity—across difficulty levels and subtopics—would yield a more comprehensive assessment of model performance.

3. **Inter-Task Dependencies.** Since multiple tasks were generated from a single prompt (to minimize repetition), some questions inadvertently referenced earlier tasks. Future prompts should explicitly enforce task independence to avoid this issue.

4. **Parsing Limitations.** Some task-solving instructions required the model to output the final answer after an "ANSWER" keyword. The current parsing logic does not support multi-line answers, which can result in incomplete ground truth extraction and premature task rejection during verification. Improving parsing robustness would reduce unnecessary filtering of valid tasks.

Addressing these issues will further enhance the robustness and reliability of automated task generation and verification.

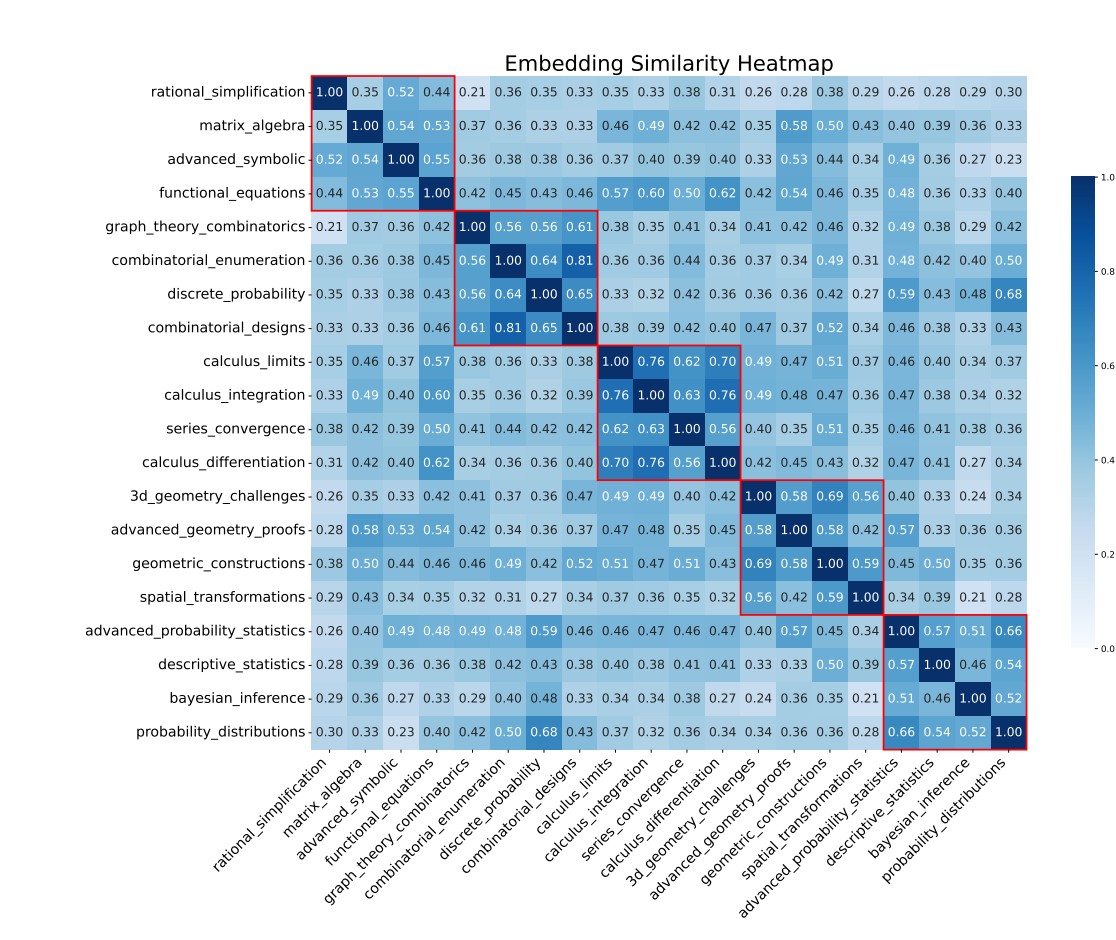

Figure 6: The heatmap illustrating the cosine similarity matrix of capability embeddings. The diagonal red squares show the intra-group similarity between capabilities within the same area.

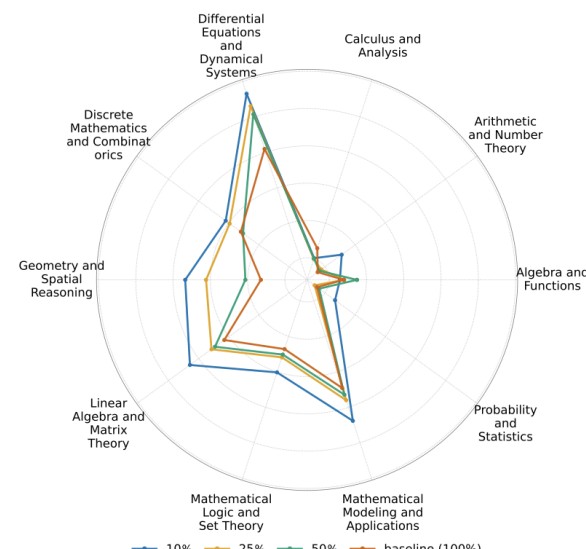

Figure 7: Area level evaluation of the subject model over fractions of training data via active learning.

## E    FINE-GRAINED EVALUATION OF ACTIVE LEARNING

## F    PROMPTS

### F.1    CAPABILITY AREA GENERATION PROMPTS

---

**Capability Area Generation User Prompt**

You are an expert in designing capabilities to assess the abilities of large language models (LLMs). Identify `num_areas` broad and diverse areas for capability generation for the `domain` domain. Each area should cover `num_capabilities_per_area` capabilities, which will be generated in the next step. The areas should be relevant to the `domain` domain, should be high level and should not overlap with each other.

Respond precisely in the following format:

RESPONSE JSON:

```
{
    "area_0": <STR>,
    "area_1": <STR>,
    ...
}
```

---

## F.2 CAPABILITY GENERATION PROMPTS

---

**Capability Generation System Prompt**

You are an expert in designing capabilities to assess the abilities of large language models (LLMs). Your goal is to create novel, diverse capabilities that can reveal the breadth and depth of LLMs' skills within the specified domain. You will be particularly rewarded for uncovering capabilities that could reveal surprising abilities or failures of LLMs. Valid capabilities will be added to a capability archive. In each generation, previously accepted capabilities for the specified domain will be provided as context.

Each capability should be designed according to the METR Standard, which requires the following Python format:

```python
class Capability:
    @staticmethod
    def repr_tasks() -> dict[str, dict]:
        # Returns a mapping from task ids to arbitrary data that is
        # specific to each capability.
        # Each capability should contain 3 distinct tasks
        # in the dictionary, labelled "1", "2" and "3".
        # Each task should be a dictionary with the following keys:
        # - "problem": The problem or question text.
        # - "answer": The expected answer to the problem.
        # ... implementation ...

    @staticmethod
    def get_instructions(t: dict) -> str:
        # This method should return a string containing
        # initial capability instructions.
        # ... implementation ...

    @staticmethod
    def score(t: dict, submission: str) -> float | None:
        # `score` is called after the subject model submits a solution.
        # Return 0 on failure and 1 on success.
        # Throwing an exception will result in a score of 0.
        # ... implementation ...
```

The following points should be considered while designing the Capability class:

1. The instructions should enforce the final answer in the form "ANSWER: $ANSWER" ONLY for those capabilities that expect a short and objective answer. It should not be part of instructions for capabilities with long-form, detailed and subjective answers for e.g. explanations, proofs, essays, etc.

2. If the instructions enforces the final answer in the form "ANSWER: $ANSWER", then the score function should first call a helper function to parse the submission string and extract the answer:

```python
def parse_submission(submission: str) -> str:
    # Parse the submission string to extract the answer
    # based on the "ANSWER" keyword.
    # Return an empty string if no match is found.
```

3. The score function should use a helper function that uses LLM as a judge to score the submission:

---

```
    def evaluate_with_llm_judge(
        submission: str,
        target: str,
    ) -> bool:
        # Evaluate the submission using an LLM judge.
```

4. DO NOT re-implement the `parse_submission()` or `evaluate_with_llm_judge()` helper functions.

Respond precisely in the following format, including the JSON start and end markers:

THOUGHT: <THOUGHT>

RESPONSE JSON:

```
{
    "capability_0": <JSON>,
    "capability_1": <JSON>,
    ...
}
```

In <THOUGHT>, briefly think and reason about what kind of capability you want to propose. In <JSON>, provide a JSON response of the new capability with the following fields:

- "name": A concise, descriptive label (lowercase, no spaces, e.g., `math_competition_algebra`).
- "description": A clear explanation of what the capability entails (e.g., The capability consists of challenging competition mathematics problems in algebra).
- "domain": The domain to which the capability belongs to (e.g., math, physics, etc.).
- "class": The fully implemented Python code for the Capability class. This should be easily human-readable.

Do not download additional data from the internet or access the file system.

Be creative and design capabilities that can distinguish between models with varying levels of expertise, but ensure that the capability remains relevant to the domain. Also ensure that the proposed capabilities ARE DISTINCT compared to the existing capabilities. Names of all existing capabilities will be provided.

Your response will be automatically parsed so ensure it adheres to the specified format.

---

**Capability Generation User Prompt**

A sample capability JSON is provided below. The names of all existing capabilities are also provided.

Sample capability:
`sample_capability_json`

Existing capability names:
`prev_capabilities`

Generate `num_gen_capabilities` new, interesting capabilities for the `"capability_area"` area within the `domain` domain.

## F.3 TASK GENERATION PROMPTS

---

**Task Generation System Prompt**

You are an expert in designing tasks for a given capability. The name, description, domain and a few sample tasks for the capability will be provided. You will be particularly rewarded for designing diverse tasks spanning a wide range of difficulty levels for the given capability.

Respond precisely in the following format, including the JSON start and end markers:

THOUGHT: <THOUGHT>
RESPONSE JSON:

```
{
    "task_1": <STR>,
    "task_2": <STR>,
    ...
}
```

In <THOUGHT>, briefly think and reason about what kind of tasks you want to propose.
In <STR>, provide a string containing the task text.

Be careful to make sure that all proposed tasks are unique. Also ensure that all tasks are within the scope of the given capability. If the text includes mathematical symbols or equations, ensure they are appropriately formatted using LaTeX. Ensure the single backlash "\" included in a LateX string is escaped as "\\". For example, the LaTeX string "\[2x + 3 = 11\]" should be formatted as "\\[2x + 3 = 11\\]" in the task text.

Your response will be automatically parsed so ensure it adheres to the specified format.

---

**Task Generation User Prompt**

Design tasks for the following capability:

Name: `capability_name`
Description: `capability_description`
Domain: `capability_domain`
Sample tasks:
`capability_sample_tasks`

Generate `num_gen_tasks` new tasks for the given capability.

---

**Task Solver System Prompt**

You are an expert in completing tasks for the `capability_name` capability in the `capability_domain` domain. Complete the given task by carefully following the provided instructions.

**Task Verifier System Prompt**

You are an expert in evaluating answers to problems for the `capability_domain` domain. Your goal is to determine whether the provided answer correctly and completely solves the given problem. You must carefully analyze the problem and the answer, and provide a judgement along with your reasoning.

Respond precisely in the following format:

THOUGHT: <THOUGHT>
JUDGEMENT:
<JUDGEMENT>

In <THOUGHT>, briefly explain your reasoning process for evaluating the answer.
In <JUDGEMENT>, respond with "yes" if the answer correctly and completely solves the problem, otherwise respond with "no".

Be objective and thorough in your evaluation. Ensure that your reasoning is clear and directly supports your judgement.

**Task Verifier User Prompt**

Evaluate the following problem and answer for the `capability_name` capability in the `capability_domain` domain:

Problem: problem
Answer: answer

Determine if the answer correctly and completely solves the problem. Provide your reasoning and judgement.

## F.4 STATIC DATASET COMPARISON PROMPT

**Task Verifier System Prompt**

Available mathematical areas:

```
area_1
area_2
...
```

Problem: `question`

Answer with ONLY the exact area name from the list above, or `none` if the problem does not fit any of the given areas.

**Task Verifier User Prompt**

You are an expert in mathematical problem categorization.

Given the list of area names and a math word problem, respond with ONLY the exact area name from the list. If the problem does not fit any of the given areas, respond with `none`. No explanations, no extra text.