# OpenReview forum: "Automated Capability Evaluation of Foundation Models"
_ICLR.cc/2026/Conference — ICLR 2026 Conference Withdrawn Submission_

### Official Review · Reviewer_dykv · 2025-10-28

**Soundness:** 2
**Presentation:** 3
**Contribution:** 2
**Rating:** 2
**Confidence:** 4

**Summary:**

This paper introduces the Active learning for Capability Evaluation (ACE) framework, a novel approach designed to address the limitations of static, human-curated LLM benchmarks. The authors propose reframing evaluation as the approximation of a latent capability function which maps model skills across a semantic space. The ACE framework operates in two stages. The first is Automated Capability Discovery and Task Generation. A "Scientist Model" (a SOTA LLM) is used to recursively decompose a domain (demonstrated with Mathematics) into a hierarchical map of capabilities. The same model will generate a large volume of problems and verifiable solutions for each capability. In the Mathematics domain, ACE generated 433 capabilities and 11,800 tasks, claiming to cover 94% of Wikipedia-defined skills. The second is Principled Efficient Evaluation. The authors employ Gaussian processes and active learning to select a minimal and most informative subset of capabilities for testing. This method significantly reduces the required evaluation budget. The authors compared results on their benchmark to the fine-grained diagnostic maps against aggregate scores from existing benchmarks, i.e., MATH, GSM8K, demonstrating the framework's potential for providing deeper, more actionable insights into model performance differences.

**Strengths:**

Originality: While fine-grained hierarchical evaluations and synthetic task generation have been explored, the paper's original contribution is the formulation of LLM evaluation as an active learning problem on a latent capability function. The two-stage framework, which combines LLM-driven knowledge decomposition and task synthesis with Gaussian processes and active learning, is a powerful and original approach for scalable diagnostics.

Clarity: The paper is well-structured, and the methodology is technically sound. The paper successfully demonstrates good diagnostic coverage and shows that near-optimal prediction of the full capability score (within 0.01 RMSE) can be achieved by evaluating fewer than 50% of the tasks, effectively validating the efficiency claim.

Significance: The primary significance of ACE lies in its efficiency and scalability. Unlike static, high-difficulty benchmarks which test aggregate skill, ACE provides a granular diagnostic map. The use of active learning makes dynamic evaluation practically viable, offering researchers a pathway to rapidly generate and test hypotheses about model weaknesses without relying on fixed, saturated benchmarks.

**Weaknesses:**

1. The ceiling effect is a critical methodological flaw. The diagnostic map's resolution is limited to the current frontier model's knowledge, resulting in an unreliable capability profile for truly advanced, research-level concepts. For domains as serious as mathematics, the lack of human audit and curation at the high end means that the generated tasks cannot capture the complexity and novelty found in expert-created datasets like FrontierMath, HARDMath or DeepMath-103K. The framework, therefore, cannot diagnose the model's capacity for the "unknown unknowns" necessary for scientific breakthroughs, undermining its utility as a frontier evaluator.

2. The paper lacks validation for the semantic capability space and tree structure. The efficacy of the Gaussian process and active learning hinges entirely on the quality and validity of the latent semantic space and the capability tree structure generated by the scientist model. The paper does not provide sufficient evidence or human validation that the LLM-derived hierarchy (e.g., how the 433 capabilities are grouped) accurately reflects the structure of human mathematical knowledge or difficulty. If the initial semantic embeddings used to define the latent space are flawed, the entire premise breaks down, meaning the claimed efficiency and diagnostic accuracy are built on an unverified structural assumption.

3. The paper has a limited scope with untested generalizability. The entire empirical validation is restricted to Mathematics, a domain with objective, deterministic ground truths. The critical claim that ACE can extend to subjective domains (like creative writing or abstract reasoning) using a "Judge Model" is completely untested. In subjective domains, the challenge of mitigating judge model bias, hallucination, and controlling for inter-rater disagreement is substantially higher, casting significant doubt on the generalizability of the claimed efficiency gains and diagnostic power.

**Questions:**

1. Given the diagnostic ceiling set by the scientist model and the use of saturated benchmarks (MATH, GSM8K) for comparison, can the authors propose a hybrid approach? Specifically, how can the active learning component be used to efficiently predict performance against the true performance frontier by embedding and evaluating against high-difficulty, human-curated datasets like HARDMATH, DeepMath-103K, and FrontierMath?

2. The active learning approach assumes a smooth latent capability function. How is this assumption validated or made plausible for non-deterministic, subjective domains (e.g., ethics, creativity)?

3. Why is the 180 task human audit sufficient to guarantee the claimed precision across the entire 11,800 task benchmark? Furthermore, can the active learning component be adapted to efficiently select the most uncertain tasks for human auditing, thereby maximising the return on investment for verification?

---

### Official Review · Reviewer_5bUr · 2025-10-31

**Soundness:** 2
**Presentation:** 1
**Contribution:** 2
**Rating:** 2
**Confidence:** 4

**Summary:**

This paper proposes ACE framework for evaluating the capabilities of foundation models by decomposing a domain into a hierarchy of capabilities and generating related tasks. It embeds the capabilities into a latent space and approximates a latent capability function for a model, aiming to estimate the model's performance across the capabilities.

**Strengths:**

The paper proposes a pipeline to automatically evaluate the capabilities of a model on specific domains, avoiding curating static benchmarks manually.

Moving beyond static benchmarks to an adaptive process for evaluation is considerable.

**Weaknesses:**

Limited Scope: while the paper mentions evaluating close-ended, deterministic, and open-ended tasks (Section 2.2), the experiments are confined to mathematics, a domain with highly verifiable outcomes. The framework's applicability to other domains remains unexplored.

Unjustified Comparisons: the critique of the GSM8k benchmark for lacking coverage of advanced topics like differential equations is misplaced, as its scope is intentionally limited to grade-school mathematics. Furthermore, the claim to be the "first that formulates evaluation as approximating a latent capability function" overstates its novelty, as similar approaches involving latent space embeddings for model capabilities have been widely explored in model routing.

Inherent bias: using a "scientist model" for both task generation and solution risks introducing the model's own biases. The complexity of this dual role also calls into question the generalization, since making judgments is typically simpler than generating solutions. It is possible for using weaker model to judge stronger models, but it is unreliable to use weaker models as scientist model. The reliance on manual verification further compromises the method's scalability.

Formatting Issues: there are some notable formatting problems, such as excessive spacing at the bottom of the pages.

**Questions:**

See weaknesses.

---

### Official Review · Reviewer_gYQz · 2025-11-01

**Soundness:** 2
**Presentation:** 2
**Contribution:** 2
**Rating:** 4
**Confidence:** 4

**Summary:**

The paper proposes Active Learning for Capability Evaluation (ACE), a framework that leverages LLMs to generate diverse tasks within a structured capability hierarchy for evaluating foundation models. Results demonstrate that the generated tasks achieve high coverage of Wikipedia-defined mathematical skills and relevant datasets.

**Strengths:**

- The problem of LLM capability evaluation using model-generated tasks is an important and timely topic.
- The formulation of a capability hierarchy, along with the analysis of coverage relative to Wikipedia and relevant datasets, is novel and interesting.

**Weaknesses:**

1. A major concern is the lack of clear referencing to prior work. This paper seems to build heavily on Automated Capability Discovery (Lu et al., 2025), sharing many motivations and methods with ACD. While this is fine, it should be explicitly stated in key sections like the introduction, discussing why ACD is insufficient for the research motivation, which algorithmic components are adopted, and what novel contributions the authors introduce.

2. Several components lack clarity:
  - **Capability Hierarchy:** Are categories manually defined or generated automatically?
  - **Verification:** Does human inspection occur during algorithm execution or only as post-hoc analysis?
  - **Active Learning:** What is the acquisition score, why was it used here, and how is it computed? How is the "optimal" candidate selected, and what metric defines optimality?

3. The evaluation would benefit from deeper insights into the process. The current presentation makes it difficult for readers to understand the tasks generated by the "scientist" model or interpret the reported performance numbers.

4. Appendix D briefly addresses LLM judge reliability for generated tasks, but this is a central challenge in the field and deserves more extensive discussion.

5. Most files in the anonymous GitHub repository are inaccessible.

**Questions:**

See above weakness

---

### Official Review · Reviewer_QK8i · 2025-11-02

**Soundness:** 3
**Presentation:** 3
**Contribution:** 2
**Rating:** 6
**Confidence:** 2

**Summary:**

The paper proposes ACE (Active Learning for Capability Evaluation), a framework that automates fine-grained model evaluation by (i) using frontier “scientist models” to generate a hierarchical capability space and corresponding tasks with reference solutions, and (ii) learning a latent capability function over capability embeddings, enabling active selection of which capabilities to probe for a given subject model. In Mathematics, ACE constructs 433 capabilities and 11,800 tasks, reports 94% coverage of Wikipedia-defined math capabilities, and shows that evaluating <50% of capabilities suffices to approximate the full capability function within 0.01 RMSE via Gaussian-process-based active learning.

**Strengths:**

- Well-scoped problem framing and clear pipeline. The paper cleanly reframes evaluation as approximating a latent capability function, then operationalizes it with a practical, modular pipeline (Fig. 1). This alignment of concept and system is strong.

- Large, balanced capability set with automated tasking. In math, ACE builds 433 capabilities and 11.8k tasks; distributional comparisons show balanced area coverage relative to GSM8K and MATH, which are skewed. (Fig. 2a.)

- Fine-grained comparative profiles across models. Area-level radar plots show nuanced differences across four subject models that aggregate metrics would miss (Fig. 3a).

**Weaknesses:**

- Wikipedia <-> ACE mapping relies on an LLM classifier; no human adjudication or error bars are reported.

- The AL ablation fits f on o3-mini only; it’s unclear if the sample-efficiency curve transfers across models.

**Questions:**

- Does the active-learning query trajectory depend heavily on the subject model, or are there shared informative capabilities? Can you provide any cross-model transfer results?

- Which GP kernel was used, and how sensitive are results to ARD vs. RBF vs. dot-product?

---

### Note · Authors · 2025-11-14

I have read and agree with the venue's withdrawal policy on behalf of myself and my co-authors.